# Development of a Passive Back-Support Exoskeleton Mimicking Human Spine Motion for Multi-Posture Assistance in Occupational Tasks

**DOI:** 10.3390/biomimetics10060349

**Published:** 2025-05-27

**Authors:** Jiyuan Wu, Zhiquan Chen, Yinglong Zhang, Qi Zhang, Xingsong Wang, Mengqian Tian

**Affiliations:** 1School of Mechanical Engineering, Southeast University, Nanjing 211189, China; wujy@seu.edu.cn (J.W.); zhiquanchen@seu.edu.cn (Z.C.); 220210349@seu.edu.cn (Y.Z.); tianmq@seu.edu.cn (M.T.); 2School of Automation, Nanjing University of Information Science and Technology, Nanjing 210044, China; qizhang_mech@nuist.edu.cn

**Keywords:** human spine, forward bending, biomechanical analysis, back-support exoskeleton, motion assistance

## Abstract

Passive back-support exoskeletons commonly employ elastic components to assist users during dynamic tasks. However, these designs are ineffective in providing sustained assistance for prolonged static bending postures, such as those required in surgery, assembly, and farming, where users experience continuous lumbar flexion. To address this limitation, a novel passive back-support exoskeleton inspired by the human spine is proposed in this work. The exoskeleton integrates a five-bar linkage mechanism with vertebrae-mimicking units, allowing for both dynamic flexion–extension movements and rigid support at various flexion angles. During the experiments, subjects are instructed to perform a 30-min forward-bending assembly task under two conditions: with and without wearing the exoskeleton. Compared to the free condition, the electromyography results indicate a 10.1% reduction in integrated EMG (IEMG) and a 9.78% decrease in root mean square (RMS) values of the erector spinae with the exoskeleton. Meanwhile, the metabolic rate is decreased by 11.1%, highlighting the effectiveness of the exoskeleton in mitigating muscle fatigue during prolonged static work. This work provides a promising solution for reducing musculoskeletal strain in occupations requiring sustained forward bending, making it a valuable advancement in passive exoskeleton technology.

## 1. Introduction

Lower back pain, a primary contributor to work-related musculoskeletal disorders, is frequently exacerbated by prolonged forward-bending postures required in various occupations [1]. Fruit tree farmers are at risk for musculoskeletal disorders, particularly in the back and shoulders [2,3]. Similarly, surgeons often maintain static forward-leaning positions during long-duration procedures, leading to chronic spinal discomfort and postural fatigue [4,5]. Industrial laborers involved in assembly lines and manual labor also experience high rates of lower back strain due to non-neutral postures and repetitive motions [6,7,8,9]. The high physical demands and sustained muscle activation in these tasks emphasize the urgent need for effective ergonomic interventions to alleviate back muscle fatigue and reduce the risk of long-term spinal injuries.

To address these issues, various back-support exoskeletons have been developed to assist the lumbar region, including active and passive systems [10,11]. Active back-support exoskeletons, equipped with motors and actuators, provide substantial lifting assistance and adaptive support [12,13]. However, their high mass, limited battery life, and complex control systems hinder their practicality for prolonged use in real-world occupational settings. Additionally, these exoskeletons show restrictions in leg movements, reducing user comfort and adoption rates [14]. In contrast, passive systems have the advantage of being lightweight and easy to wear. This kind of back-support exoskeleton employs mechanical components such as springs and elastic elements to store and release energy, assisting users in returning to an upright posture after bending [15,16,17].

Despite their effectiveness for dynamic and repetitive tasks [18], most existing passive designs fail to provide adequate support during prolonged static forward bending, which is prevalent in occupational settings. This gap is particularly crucial as prolonged forward bending relies heavily on the erector spinae to counteract gravitational forces and maintain spinal stability [19]. Continuous muscle activation without sufficient support leads to excessive fatigue, reducing postural endurance and increasing the risk of musculoskeletal injuries. Existing passive exoskeletons, typically composed of flexible materials, fail to offer the necessary structural rigidity to ensure stable support at various flexion angles while accommodating natural movement adjustments.

To address these limitations, a novel passive back-support exoskeleton designed to deliver adjustable support across a range of flexion angles is proposed in this work. In contrast to conventional passive exoskeletons that mainly aid in returning to an upright posture, this device extends functionality by supporting both dynamic movements and prolonged static forward bending. The design integrates a spine-inspired mechanism, ensuring rigid support in static postures and adaptive flexibility for natural movement. Experimental validation demonstrates that the proposed exoskeleton effectively reduces muscle fatigue and enhances postural endurance, offering a practical solution for occupations requiring sustained forward bending.

The rest of this paper is organized as follows. Section 2 includes the motion analysis of human spine bending. Section 3 introduces the parameter selection and mechanical design of the back-support exoskeleton in detail. Section 4 represents the experimental setup and results of the exoskeleton. Section 5 compares the results and limitations of the exoskeleton. Finally, conclusions are given in Section 6.

## 2. Motion Analysis of Human Spine

### 2.1. Anatomical Mechanism

The human back is primarily composed of the spine, muscles, and the surrounding soft tissues [20], which collectively provide structural support and flexibility. The spine serves as the primary support structure for the torso, allowing for a range of motion while maintaining stability. The ribs reinforce structural integrity while shielding vital organs. Muscles control different movements of the human body through contraction and relaxation, while soft tissues offer cushioning to maintain the stability and flexibility of the lower back.

The main structure of the spine consists of thirty-three vertebrae [21], categorized into five segments: cervical (C1–C7), thoracic (T1–T2), lumbar (L1–L5), sacral (fused), and coccygeal (fused). These vertebrae form an S-shape curve, characterized by cervical lordosis, thoracic kyphosis, lumbar lordosis, and sacral kyphosis [22] (see Figure 1). The curved configuration facilitates flexion, extension, and rotational movements of the upper body, and functions as a cushion to mitigate impacts on internal organs during motion. Vertebrae are interconnected by intervertebral discs, which function as shock absorbers between the vertebrae, and by ligaments and muscles to maintain spinal alignment and support human motion [23].

Bending forward is a fundamental movement in daily life, especially in tasks such as lifting, carrying, and prolonged stooping. This motion relies on both the flexibility of the S-shaped spine and the coordination of key muscle groups, particularly the erector spinae, which play a crucial role in spinal stability. Figure 1b illustrates the components of the erector spinae, including the spinalis, longissimus, and iliocostalis. During forward bending, these muscle groups undergo eccentric contraction, gradually lengthening to regulate the controlled descent of the upper body. As the body returns to an upright position, these muscles shift to concentric contraction, enabling the torso to rise and stabilize the spine. In addition to the erector spinae, other musculature contributes to the bending motion, including the abdominal muscles, gluteals, and hip flexors. However, prolonged or repetitive bending has the potential to induce muscular fatigue, particularly in the erector spinae, increasing the risk of lower back injuries. Understanding these muscle dynamics is essential for designing exoskeletons that provide targeted support and reduce fatigue. Given that excessive strain on the erector spinae is a key factor in lower back discomfort and injury, supportive interventions are critical for mitigating fatigue and improving movement efficiency.

### 2.2. Modeling of Spine Bending

#### 2.2.1. Kinematic Data Collection

Building upon the anatomical insights of the spine, an experimental study was conducted to investigate spinal motion during forward bending and provide empirical data for exoskeleton design. The primary goal of analyzing spinal bending kinematics was to guide the mechanical design of a passive back-support exoskeleton. Accurate characterization of spinal flexion enables the exoskeleton to align with natural trunk movement and provide effective support across various postures without hindering natural trunk motion. Real-time human spinal kinematics were recorded using a motion capture system (Qualisys, Gothenburg, Sweden) with 24 Qualisys Arqus cameras, as shown in Figure 2. The study involved a healthy adult male (height: 173 cm) with no history of spinal injury. To ensure precise tracking of spinal movement, 11 reflective markers were placed on the spinous processes of 11 vertebrae: C7, T2, T4, T6, T7, T9, T12, L1, L3, L5, and S1. The subject was instructed to maintain a neutral upright posture for 5 s, with arms relaxed naturally and gaze directed forward, then perform a forward bending motion at a steady speed until reaching maximum flexion. The process was repeated three times with a 10-s rest interval between trials to minimize fatigue effects.

Although the lower limbs of the subject remained stationary during the trials, slight posterior displacement of the hips was observed as a compensatory mechanism to maintain body balance. This shift influenced the lumbar spinal marker positions and, consequently, the spinal posture measurements. To correct for this, the coordinate trajectory of S1 vertebrae was applied as the reference point, normalizing all recorded spinal motion data to eliminate translational artifacts. The averaged normalized spine trajectory across three trials in the sagittal plane is shown in Figure 3, providing a clear depiction of spinal motion during the bending motion.

As illustrated by the bending curve in the figure, all vertebrae are involved in forward flexion, though their individual contributions vary considerably. Notably, the thoracolumbar junction (T12–L1) and the lumbosacral junction (L5–S1) exhibit the highest flexion contributions. Based on medical data analysis [25], the lumbar-to-sacral flexion ratios (L/S ratio) at 30°, 60° and 90° are 1.9, 0.9 and 0.5, respectively. Since the hip joint is anatomically aligned with the upper sacrum in the sagittal plane, the lumbar-to-sacral flexion ratio can also be considered the lumbar-to-hip flexion ratio.

#### 2.2.2. Biomechanical Analysis

The lumber-to-sacral flexion ratio changes continuously during spinal flexion, making it inappropriate to model the spine as a single-link rotational mechanism. Given the anatomical distribution of muscles and bones, the erector spinae muscle attaches to the spinous and transverse processes of the cervical and thoracic vertebrae, with its lower end anchored to the sacrum. Consequently, the spinal flexion mechanism in the sagittal plane can be simplified as a two-bar linkage mechanism, as shown in Figure 4a. In this model, points *O*, *A*, and *B* represent the upper sacrum, L1 vertebrae, and T1 vertebrae, respectively. The lengths of the lumbar and thoracic spines are denoted by $l_{1}$ and $l_{2}$, respectively. The angles $\theta_{1}$, $\theta_{2}$, and α represent the angle of the lumbar flexion relative to the upper sacrum, the angle of the thoracic flexion relative to the lumbar vertebra, and the overall spinal flexion angle, respectively.

Since the number of thoracic vertebrae is approximately twice that of lumbar vertebrae, for simplicity in calculation, it can be assumed that the length of the thoracic linkage is twice that of the lumbar linkage, that is(1)$$l_{1}=0.5l_{2}$$

Based on the distribution of the lumber-to-sacral flexion ratio during forward flexion, for the convenience of subsequent calculations, the bending angles of the lumbar and thoracic linkages approximately satisfy the following conditions for 30°, 60°, and 90° of spinal flexion:(2)$$\left\{\begin{array}{ccc} \theta_{1} & =0.5\theta_{2}, & \alpha =30{}^{\circ}\\ \theta_{1} & =\theta_{2}, & \alpha =60{}^{\circ}\\ \theta_{1} & =2\theta_{2}, & \alpha =90{}^{\circ} \end{array}\right.$$

Establishing a two-dimensional Cartesian coordinate system at point *O*, the coordinates of *B* can be expressed as follows:(3)$$\left\{\begin{array}{c} x_{B}=l_{1}sin\theta_{1}+l_{2}sin\left(\theta_{1}+\theta_{2}\right)\\ y_{B}=l_{1}cos\theta_{1}+l_{2}cos\left(\theta_{1}+\theta_{2}\right) \end{array}\right.$$

The force analysis of the human spine is illustrated in Figure 4b. In this diagram, the height of the person is denoted as *L*, and body weight as *W*. The torso weight is represented by $G_{1}$, while the total weight of the head, neck, and arms is $G_{2}$. The force *T* represents the muscle tension force provided by the erector spinae, and $F_{N}$ is the support force provided by the sacrum, which is equal to the spine compression force. $l_{3}$ refers to the distance between the erector spinae force point *C* and the upper end *B* of the thoracic vertebrae. *d* represents the perpendicular distance from the hip joint to the direction of the muscle tension force in the sagittal plane. β is the angle between *T* and the thoracic rod AC, and γ is the angle between AC and OC. δ is the angle between BC and the coronal plane and ϕ is the angle between $F_{N}$ and the horizontal plane.

According to data from the national inertial parameters of the adult human body [26], the parameters depicted in the figure should meet the following conditions:(4)$$\left\{\begin{array}{cc} G_{1} & =0.4357W\\ G_{2} & =0.1709W\\ l_{1} & =0.1333L\\ l_{2} & =0.2667L\\ l_{3} & =0.1333L\\ \beta & =12{}^{\circ} \end{array}\right.$$

The condition for the human spine to maintain balance during forward bending is given by(5)$$\left[\begin{array}{ccc} F_{N}cos\phi -Tsin\left(\beta +\delta \right) & 0 & 0\\ F_{N}sin\phi -Tcos\left(\beta +\delta \right) & G_{1}+G_{2} & 0\\ Td-G_{1}x_{C}-G_{2}x_{B} & 0 & 0 \end{array}\right]\left[\begin{array}{c} 1\\ F_{N}\\ T \end{array}\right]=\left[\begin{array}{c} 0\\ 0\\ 0 \end{array}\right]$$
where the parameters can be calculated by(6)$$\left\{\begin{array}{cc} \delta & =arccos(\frac{{\left[l_{2}cos\left(\theta_{1}+\theta_{2}\right)\right]}^{2}+l_{2}^{2}-{\left[l_{2}sin\left(\theta_{1}+\theta_{2}\right)\right]}^{2}}{2l_{2}^{2}cos\left(\theta_{1}+\theta_{2}\right)})\\ d & =\sqrt{x_{c}^{2}+y_{c}^{2}}sin\left(\beta +\gamma \right)\\ \gamma & =acrcos(\frac{{\left(l_{2}-l_{3}\right)}^{2}+x_{c}^{2}+y_{c}^{2}-l_{1}^{2}}{2\left(l_{2}-l_{3}\right)\sqrt{x_{c}^{2}+y_{c}^{2}}})\\ x_{c} & =l_{1}cos\theta_{1}+\left(l_{2}-l_{3}\right)sin\left(\theta_{1}+\theta_{2}\right)\\ y_{c} & =l_{1}cos\theta_{1}+\left(l_{2}-l_{3}\right)cos\left(\theta_{1}+\theta_{2}\right) \end{array}\right.$$

For a human body with a height of 1.75 m and a weight of 65 kg, the relevant parameters can be substituted into the equilibrium matrix to calculate the spinal forces at different flexion angles. The results are presented in Table 1.

The results indicate that as the flexion angle increases, both pressure on the spine and muscle tension force increase correspondingly, leading to an increased risk of intervertebral disc and muscle injuries. However, the aforementioned analysis considers only the condition involving human body weight. In practical scenarios, lumbar injuries frequently occur under additional loading conditions. The loads can generally be categorized into the torso-borne loads, such as backpacks or medical lead aprons, and limb-carried loads, such as handheld objects. Consequently, the forces on the spine and muscles need to be analyzed for the additional loads in both cases.

For torso-borne loads, as in the case of physicians wearing lead aprons, the center of mass of the lead apron can be considered to coincide with that of the torso. Thus, the mass of the lead apron (5 kg) is added to the torso mass $G_{1}$. Recalculating the matrix, it can be found that when the flexion angle reaches 90°, the spinal compression force reaches 1141 N and the muscle tension force increases to 1150 N. For limb-carried loads, such as workers holding welding tools, the weight of the welding tools (2 kg) is added to $G_{2}$, resulting in the pressure on the spine of 1022 N and the muscle tension force of 1027 N at the same flexion angle.

These findings highlight that even slight increases in load relative to body weight may cause a marked rise in spinal pressure, underscoring the importance of load management. For manual laborers who sustain prolonged forward-bending postures, the significant increase in pressure and shear force may cause disc herniation, nerve root compression, resulting in ligamentous damage, and muscle spasms.

## 3. Design of Back-Support Exoskeleton

This section presents the design of the back-support exoskeleton, which is derived from the spine kinematics discussed in Section 2. By analyzing the design requirements, the mechanism tracking the motion of the user backbone has been identified. Subsequently, the specific design for the exoskeleton is introduced.

### 3.1. Exoskeleton Modeling

Exoskeleton devices have the capability to reduce the burden on the human body, enhance work efficiency, and mitigate the risk of occupational injuries. Determining the maximum torque that an exoskeleton support system must withstand is a critical prerequisite before its design, serving as the foundation for subsequent parameter development. To reduce muscular strain, the exoskeleton should provide support to the erector spinae muscles in maintaining the torso’s posture when the body bends forward beyond a certain angle [27]. As discussed in the previous section, for the case of carrying an additional torso-borne load, when the body flexion angle is 90°, the tension force provided by the erector spinae and the support force provided by the sacrum are the largest. Therefore, the torque required from the exoskeleton under this case is modeled and analyzed, as shown in Figure 5.

To closely mimic the natural structure and motion of the human body, the exoskeleton is designed as a multi-bar linkage mechanism, allowing for the force transmission of the torso and extra load. To accommodate spinal motion patterns, the exoskeleton adopts a five-bar linkage mechanism to imitate the natural curvatures of the human spine. The five-bar linkage mechanism is designed to anatomically correspond to key vertebral points along the human spine. Specifically, the joints and both ends align with the vertebrae C7, T7, T12, L3, and S1, as well as the pelvis. The segment formed by C7–T7–T12 mimics the thoracic kyphosis of the human spine, with T7 serving as the apex of the curvature. Similarly, the T12–L3–S1 segment replicates the lumbar lordosis, with L3 as the apex of the curvature. The S1–pelvis segment functions as the structural base of the spine, analogous to the sacral-pelvic interface in the human body.

Here, $F_{exo}$ represents the support force provided by the exoskeleton to the human body, and $l_{4}$ is the vertical distance from the exoskeleton support force point to the upper thoracic vertebra *B*. $l_{5}$ is the vertical distance from the first joint of the exoskeleton to the hip joint in the sagittal plane. $J_{i}$ denotes the joint torque at the *i*-th joint of the exoskeleton, and ΔG represents the external load weight.

For the exoskeleton to maintain torque balance at point *O* during static support, the conditions $\sum M_{O}=0$ should be satisfied, namely(7)$$\begin{array}{cc} \; & F_{exo}\left[l_{1}sin\theta_{1}+l_{2}sin\left(\theta_{1}+\theta_{2}\right)-l_{4}\right]=Td \end{array}$$
Therefore, the maximum torque that the exoskeleton should provide can be expressed as(8)$$J=F_{exo}\left[l_{1}sin\theta_{1}+l_{2}sin\left(\theta_{1}+\theta_{2}\right)-l_{4}+l_{5}\right]$$

From the above equations, it can be observed that the support force provided by the exoskeleton is constantly changing with the different flexion angles of the exoskeleton. The maximum support force that the exoskeleton can provide determines the maximum load-carrying capacity of the exoskeleton. Subsequently, the parameters and workspace of the five-bar linkage mechanism of the exoskeleton are analyzed. Studies indicate that the average increase in spinal length after forward flexion for a normal adult is typically 1.6 cm [28]. Therefore, the five-bar linkage mechanism adopts a recurved structure to increase the length by transforming from an S-shape to a C-shape, as depicted in Figure 5b.

Although individual variations exist in spinal curvature [29], the range of spinal angle rotations remains within a constrained range. J.J. Vacheron [30] calculated the changing angles between the intersecting segments during standing and bending by marking the positions of the spinous processes of two adjacent segments. Based on national standards for body segments division and the research of J.J. Vacheron, key parameters for the exoskeleton linkage system are specified, as presented in Table 2.

The lengths of the five-bar linkage segments were determined based on the intersegmental distances among six anatomical points (C7, T7, T12, L3, S1, and pelvis) obtained from prior human trunk flexion experiments. This configuration aims to replicate the geometric structure of the human spine. The initial angles between the linkages were also defined according to the intervertebral angles formed by these six points in the upright posture. During flexion, changes in linkage angles were determined based on the dynamic variation in positions and intersegmental angles of these anatomical landmarks, allowing the mechanism to mimic spinal motion throughout the bending process.

Given the parameters of each linkage, the workspace of the five-bar linkage mechanism is calculated via the D-H parameter method. Subsequently, the motion range of the end effector *P* is verified, and the results are shown in Figure 6, where the green point cloud represents the workspace of end effector *P* in the sagittal plane. The stacking of the flexion trajectory of the human spine in the sagittal plane clearly demonstrates that the motion range of the linkage end nearly fully covers the flexion trajectory of the human spine, thus confirming that the selected parameters fulfill the design requirements of the exoskeleton.

### 3.2. Structural Description

The overall mechanical design of the exoskeleton, illustrated in Figure 7, primarily comprises a back-support structure and a lower-limb support structure, which facilitate load transfer to the ground while enabling the user to maintain flexibility in various bending postures.

The back-support structure of the exoskeleton primarily consists of an adaptive spine-like mechanism and a pulley steering mechanism, as shown in Figure 7. The main frame of the spine-like structure is composed of vertebral units connected via linking rods. Each vertebral unit contains a ratchet mechanism, a compression spring, and a limit mechanism. The ratchet is affixed to the linking rod, while the pawl, connected to the compression spring, engages with the ratchet. A cable is connected to the pawl, routed through the pulley steering mechanism, and finally collected by the rope retraction unit. When the cable is pulled downward, disengaging the pawl from the ratchet, allowing free rotation of the ratchet and linking rod, thereby resulting in flexion and extension along with the human body. When the cable is released, the compression spring re-engages the pawl with the ratchet, locking the vertebral units in place and providing rigid support and stability to the posture of the user. This mechanism enables the wearer to perform dynamic movements, replicating the natural curvature and the alignment of the human spine, while providing enough rigidity to support the upper torso and additional loads under static conditions.

The lower limb support structure consists of hip, knee, and ankle joints, along with rigid linkages, primarily responsible for adjusting the stiffness of the back-support structure and providing direct ground support to the upper torso and extra loads. The linkages between joints are designed as telescopic rods, allowing for minor adjustments to accommodate users with different body shapes. The stiffness adjustment mechanism is embedded in the hip joint and can be adjusted quickly manually. It consists of a cable retraction unit and a metamorphic mechanism. In the cable retraction unit, the cable is wound around a torsion spring, which provides continuous tension to the cable. When bending forward, the cable drives the torsion spring to rotate and store elastic potential energy. During extension, the cable retracts and the elastic potential energy is released, helping the body return to an upright posture. The metamorphic mechanism controls cable movement to adjust the stiffness of the back-support structure. When the upper body needs to bend and stretch flexibly, the manual lever of the metamorphic mechanism is rotated upward clockwise to push the connecting rod, clamping the cable and moving it downward clockwise. The cable drives the pawl to disengage the ratchet, and the upper back support structure can flexibly follow the dynamic movement of the human body. When static work is required, just loosen the manual lever, and the manual lever can return to the initial position under the action of the spring and release the cable. At this time, the pawl re-engages the ratchet, and the rigidity of the back support structure is achieved. Due to the characteristics of the ratchet, the rigid support of the exoskeleton is only for flexion movements. When extension is required, it can be directly extended freely without any operation. The knee joint adopts a cross four-bar linkage mechanism to simulate the variable instantaneous center of rotation of the human knee joint. This ensures kinematic compatibility between the exoskeleton and the natural knee motion of the user, thereby minimizing interference with movement. Additionally, a random stop hinge is incorporated on the outer of the cross four-bar linkage, enabling the knee joint to lock at any flexion angle, thereby providing rigidity when necessary. The ankle joint is designed with an automatic clamping mechanism, enabling quick and secure attachment to the calf and ankle joints, making it easy to put on and take off. The lower limb support structure, with its integrated hip, knee, and ankle mechanisms, provides rigid support during both standing and flexed postures.

## 4. Experiments and Results

### 4.1. Experimental Conditions

Based on the proposed design scheme, a prototype of the back-support exoskeleton was fabricated and subjected to experimental validation. The exoskeleton prototype was modeled and designed using SolidWorks 2020 (Dassault Systèmes, Paris, France). During fabrication, the main structural frame was constructed from aluminum alloy to ensure both strength and lightweight performance. The back plate was manufactured using 3D-printed resin, providing a lightweight yet ergonomic interface with the user torso. The vertebral units were constructed using a combination of carbon fiber plates and stainless steel for structural durability. The total mass of the exoskeleton is 3.2 kg, making it suitable for extended wear in occupational settings without introducing excessive load to the user. The validation experiments were categorized into two aspects, including prototype stability performance tests and user performance evaluation tests, as shown in Figure 8.

For the prototype stability performance tests, the exoskeleton was suspended from an aluminum frame, and inertial sensors (IMU900) were installed on the exoskeleton to measure its posture changes. The exoskeleton was adjusted to three predefined flexion angles (30°, 60°, and 90°), and then loading was applied via a motor-driven cable attached to the back plate. Continuous loading was controlled by the motor and the full-loaded data were calculated based on Equations (2)–(8), which are 16 kg, 22.5 kg, and 30 kg, respectively. The motor was programmed to apply loads incrementally from 0 to 100% of the full load for each flexion condition. Real-time data from the inertial sensors were recorded throughout the loading process, and each loading scenario was repeated three times per flexion condition to ensure consistency.

For the user performance tests, six participants with similar body shapes and no history of lower back injuries were selected (age: 25; weight: 70 ± 5 kg). All human subjects studies were conducted following the principles of the Declaration of Helsinki. Consent was obtained from the subjects after the details and possible consequences of the study were explained. The experimental conditions were divided into two groups: performing tasks with the back-support exoskeleton (With Exo) and performing tasks without the back-support exoskeleton (No Exo). The order of the two experimental conditions was randomized for each subject. The subject first performed five repetitions of deep flexion movements without wearing the exoskeleton, followed by an assembly task under both experimental conditions. During the task, brief upright rests were allowed, but the assembly task was required to be completed within 30 min. After completing the task, participants again performed five repetitions of deep flexion movements without the exoskeleton, followed by a one-hour rest before the next trial. Each condition was repeated three times for every participant.

Electromyography (EMG) signals of the erector spinae muscles were measured (Delsys Trigno Avanti Sensor, MA, USA) during deep flexion movements performed before and after the assembly tasks. Subsequently, the measurement data were processed through filtering. The sampling frequency was set to 1024 Hz. To eliminate low-frequency noise such as motion artifacts and electrocardiographic interference, a band-pass filter (30–350 Hz) was applied. To compare the muscle fatigue levels with and without exoskeleton assistance, EMG signals of the erector spinae were analyzed to obtain the integrated EMG (IEMG) and root mean square (RMS) values.

The metabolic rate was measured during the assembly tasks with a portable gas-exchange measurement device (Cosmed K5, Rome, Italy). Brockway’s standard equations were employed to process data. Post-hoc paired *t*-tests were conducted to identify significant changes in metabolic cost, with a significance level set at 0.05 for all analyses.

### 4.2. Experimental Results

The stability performance of the back-support exoskeleton under varying flexion angles is shown in Figure 9. The relationship between the forward flexion angle changes and the loading ratio (0–100%) was systematically analyzed. With the exoskeleton initially configured to flex at 30°, 60°, and 90° under unloaded conditions, the forward flexion angle was observed to increase progressively with higher load ratios. Under full load, the flexion angle increased by 2.6°, 3.7°, and 4.3°, respectively. In addition, the root mean square error (RMSE) values remained below 3° across all tested conditions, with minimal variation observed between flexion angles (30°: 1.8°; 60°: 2.1°; 90°: 2.7°). These results demonstrate the reliability of the exoskeleton in maintaining stability under dynamic loading conditions.

As illustrated in Figure 10, the physiological impact of exoskeleton-assisted work on human ergonomics was evaluated through EMG analysis. Using pre-task EMG signals as a baseline, a significant reduction in erector spinae muscle activation was observed when wearing the exoskeleton compared to the No Exo condition. The IEMG decreased by 10.1% (p<0.05), while RMS declined by 9.78% (p<0.05), with both metrics exhibiting statistically significant differences between conditions.

During the 30-min task, metabolic rate was calculated for oxygen consumption during the first 10 min, the first 20 min, and the entire duration (Figure 11). The metabolic rate during exoskeleton-assisted work exhibited a significant reduction compared to the non-assisted condition. In the first 10 min, wearing the exoskeleton resulted in a 12.6% decrease in metabolic rate (p<0.05), with mean values declining from 1.66 ± 0.34 W/kg (No Exo) to 1.45 ± 0.35 W/kg (With Exo). For the entire duration, wearing the exoskeleton showed a reduction of 11.1% in metabolic rate compared to the No Exo condition. The observed metabolic savings align with the reduced muscular activation patterns shown in Figure 10, collectively demonstrating the system efficacy in enhancing biomechanical efficiency and mitigating user fatigue.

## 5. Discussion

The results demonstrate the effectiveness of the proposed exoskeleton with adaptive stiffness modulation for multi-posture support in alleviating the physiological strain experienced during trunk flexion with load carriage. By dynamically adjusting the stiffness based on the posture of the wearer, the exoskeleton successfully provides targeted support, reducing the load on the lumbar spine and associated muscles. EMG analysis indicates that the muscular activity of the erector spinae consistently decreases with the use of the passive back-support exoskeleton, aligning with findings from previous studies [18,31,32]. The IEMG value and RMS value of the erector spinae muscles decrease by 10.1% and 9.78%, respectively, compared with the non-assisted control group, indicating that the exoskeleton can effectively reduce the pressure on the back muscles. This reduction in muscle activity can be attributed to the mechanical redistribution of the external load through the exoskeleton structure. Specifically, the applied external load is partially redirected from the trunk to the lower limbs via the rigid five-bar linkage and lower limb support structure, thereby offloading the lumbar spine. When the user adopts a flexed posture, the exoskeleton locks in place with high stiffness, forming a quasi-static support that mechanically resists the gravitational torque acting on the upper body. This significantly reduces the internal moment that would otherwise be borne by the back extensor muscles, hence lowering muscle activation. Meanwhile, the metabolic rate was effectively reduced by 11.1% when wearing the exoskeleton. This reduction in metabolic rate not only indicates a decrease in energy expenditure but also suggests a potential for improved endurance during prolonged tasks. In situations where individuals are required to maintain a specific posture over extended periods, such as in manual labor or medical professions, preserving body energy becomes critical for improving performance and preventing fatigue. From a physical perspective, the reduction in metabolic cost is closely related to the decreased requirement for active muscular stabilization. As the exoskeleton provides external structural support, the need for continuous neuromuscular control is reduced, thereby lowering the energy consumed by muscle fibers for force generation and postural maintenance.

The back-support exoskeleton designed in this study differs from many existing exoskeletons that primarily focus on dynamic tasks such as repetitive lifting and carrying. Instead, the proposed exoskeleton is specifically designed to alleviate lower back muscle fatigue during prolonged static forward-bending postures, which are common in labor work. A key feature of the proposed exoskeleton is the integration of a five-bar mechanism with a metamorphic structure, enabling the exoskeleton to lock at any flexed posture while maintaining high rigidity. This mechanism of fixation at various flexion angles ensures continuous support for the lower back of the user, thereby reducing the sustained muscular effort required to maintain the flexion posture. Additionally, the rigid lower-limb support structure effectively transfers part of the upper body weight directly to the ground, minimizing the load borne by the back muscles. The erector spinae muscles play a critical role in the maintenance of bending posture and spinal stability. Prolonged activation of the erector spinae muscles without adequate support can lead to fatigue and discomfort, potentially resulting in chronic pain problems and lower back injuries. By providing rigid support from the exoskeleton, the required muscle activation level is significantly reduced, thereby allowing the user to maintain the intended posture with less exertion over extended periods. The EMG analysis validates the fatigue-reducing effect of the exoskeleton.

The proposed exoskeleton offers a more adaptable and ergonomic solution by allowing users to freely adjust their flexion angles. This adaptability ensures comfort and usability across different tasks and working conditions. The current version of the exoskeleton has certain limitations in terms of adaptability to users with significantly different body shapes or sizes. While the design incorporates adjustable telescopic links to accommodate minor anthropometric variations, this adjustment range may not be sufficient to ensure safe and effective support for users who are substantially taller, obese, or have atypical body proportions. In such cases, a custom exoskeleton design is necessary. The design framework proposed in this study, including the five-bar linkage modeling approach and the spinal kinematic mapping process, can be replicated and adapted for individualized customization. Future work will focus on further optimizing the structure of the exoskeleton and continuously conducting usability studies, including both quantitative scales (e.g., comfort rating) and task performance, to evaluate long-term wearability and user satisfaction in real industrial scenarios.

## 6. Conclusions

This study presents the design and evaluation of a passive back-support exoskeleton that provides multi-posture assistance, aiming to reduce muscle fatigue and metabolic cost during prolonged bending activities. To facilitate natural movement and reduce fatigue, the designed exoskeleton employs a biomimetic spinal mechanism that allows coordinated flexion and extension with the human body, while ensuring sufficient rigidity during prolonged bending postures. The experimental results demonstrate that the exoskeleton effectively reduces muscle activation of the lower back and lowers metabolic demand, indicating its potential for improving endurance and alleviating fatigue in industrial and medical applications.

This work contributes to the development of more adaptable and supportive passive exoskeletons, providing a novel approach to long-duration posture assistance and expanding the applicability of wearable assistive technologies. Future research will focus on further optimizing the stiffness modulation, improving user adaptability, and conducting real-world validations to further improve the practicality and effectiveness.

## Figures and Tables

**Figure 1 biomimetics-10-00349-f001:**
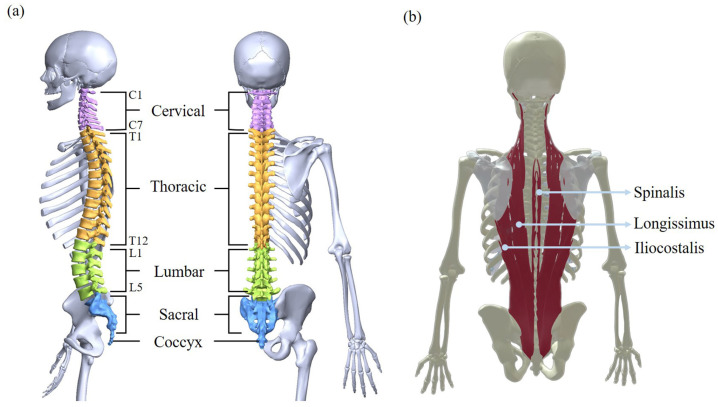
Autonomy of the human spine. (**a**) Spine-related bones. (**b**) Muscles for flexion [24].

**Figure 2 biomimetics-10-00349-f002:**
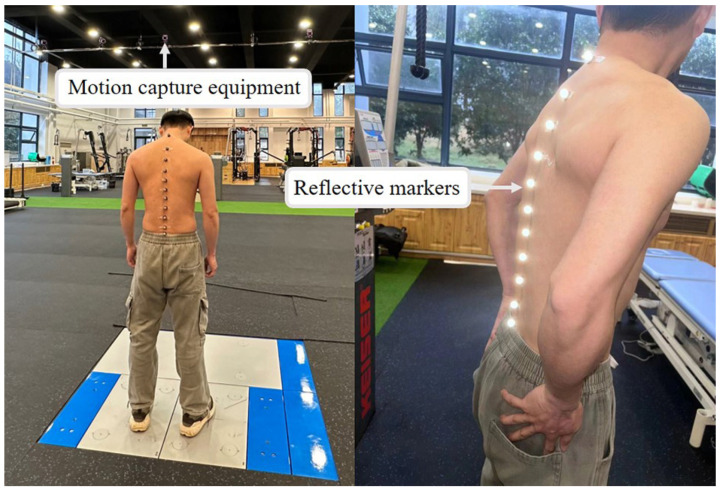
Experimental conditions for data acquisition of human spine bending motion.

**Figure 3 biomimetics-10-00349-f003:**
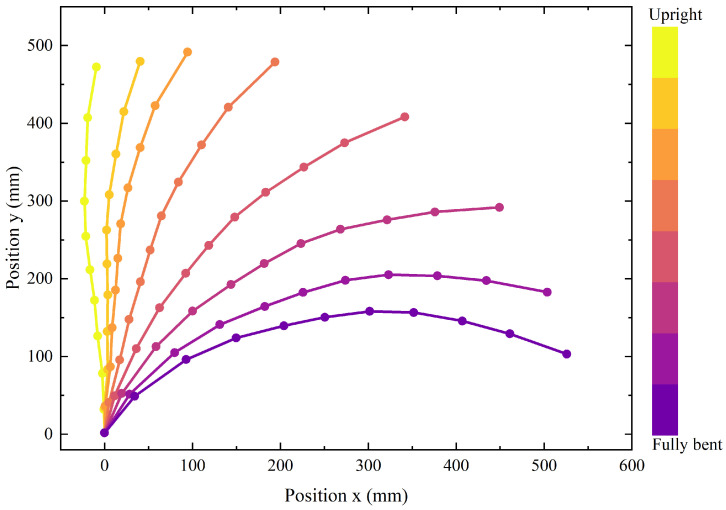
Spine trajectory during the bending motion in the sagittal plane.

**Figure 4 biomimetics-10-00349-f004:**
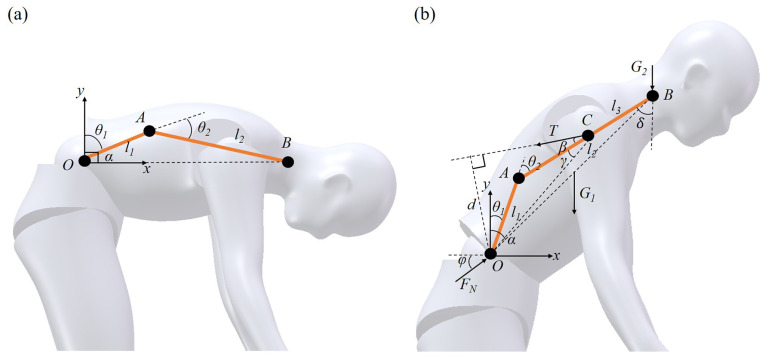
Bending model in the sagittal plane. (**a**) Two-bar mechanism model for the spine. (**b**) Force analysis of the spine when bending.

**Figure 5 biomimetics-10-00349-f005:**
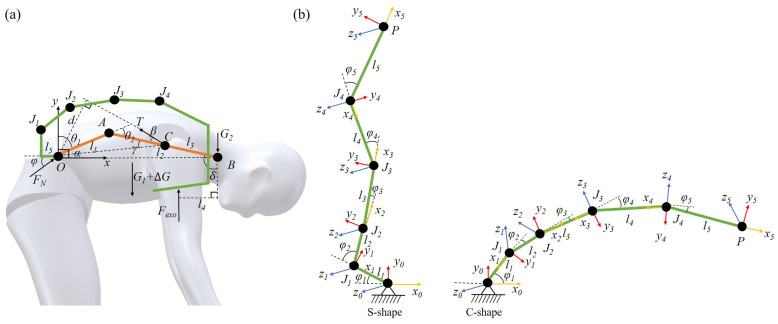
Modeling and force analysis of the back-support exoskeleton. (**a**) Design scheme of the exoskeleton. (**b**) Analysis of the five-bar linkage mechanism.

**Figure 6 biomimetics-10-00349-f006:**
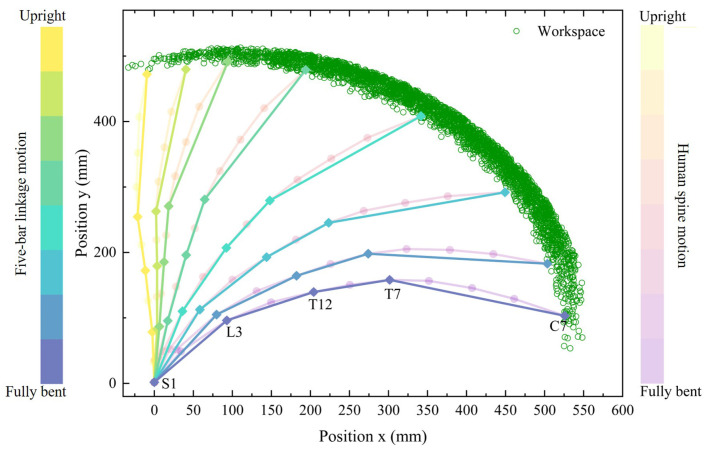
Workspace analysis of the five-bar linkage mechanism. S1-pelvis linkage $L_{1}$ is not moved during bending motion and is neglected in the figure.

**Figure 7 biomimetics-10-00349-f007:**
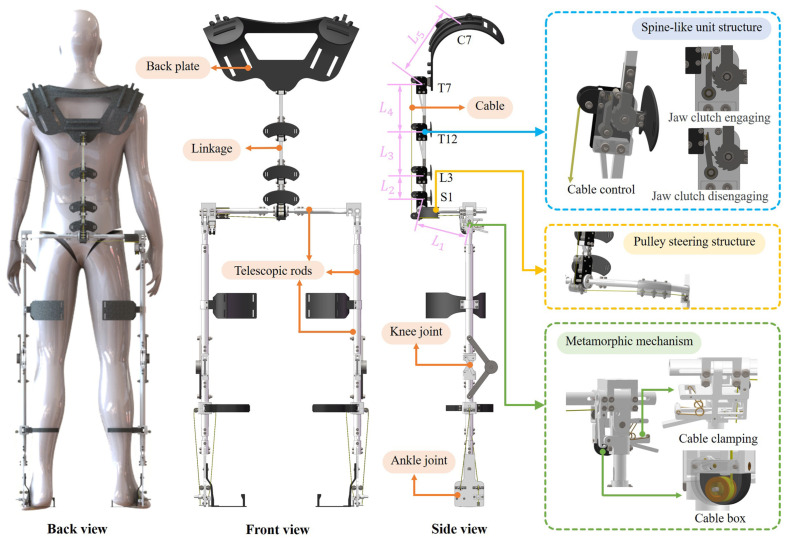
Schematic diagram of the back-support exoskeleton. The five bars of the linkage are labeled as $L_{1}$ to $L_{5}$.

**Figure 8 biomimetics-10-00349-f008:**
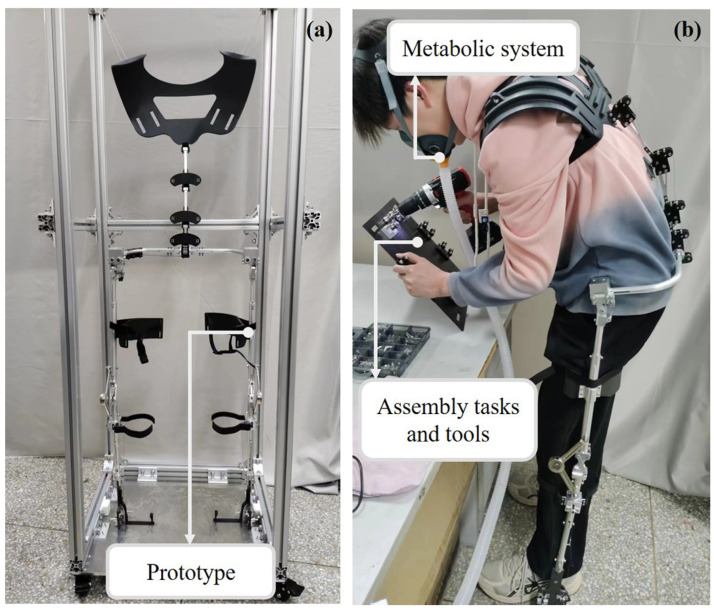
Experimental scenarios. (**a**) For prototype stability performance tests. (**b**) For the user performance evaluation tests.

**Figure 9 biomimetics-10-00349-f009:**
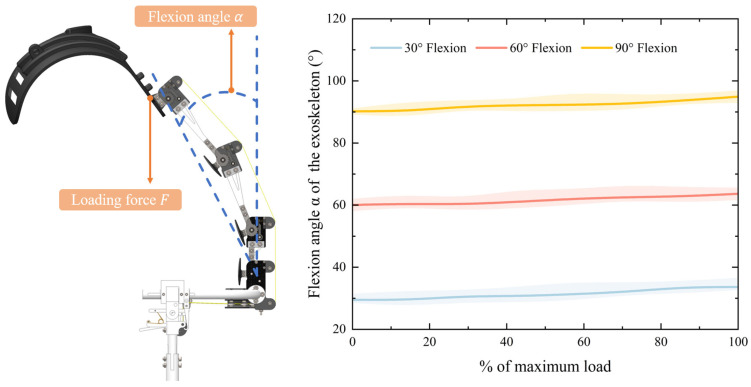
Variation in the flexion angle α of the back-support exoskeleton.

**Figure 10 biomimetics-10-00349-f010:**
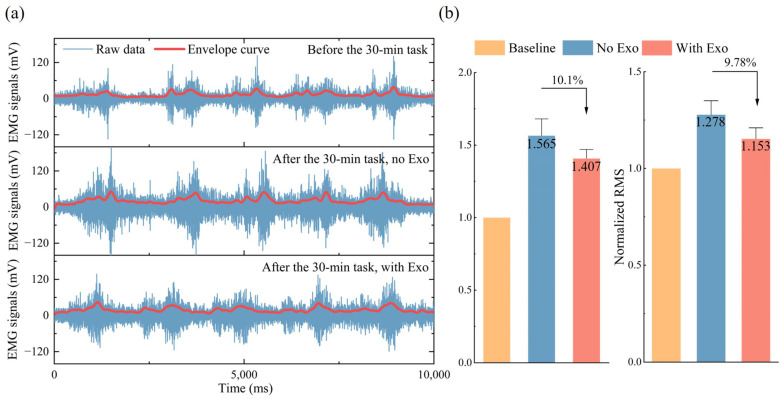
EMG performance. (**a**) EMG signal comparisons including the raw data and post-process data performance in Baseline, No Exo, and With Exo conditions, respectively. (**b**) Normalized IEMG and RMS comparisons in Baseline, No Exo, and With Exo conditions, respectively.

**Figure 11 biomimetics-10-00349-f011:**
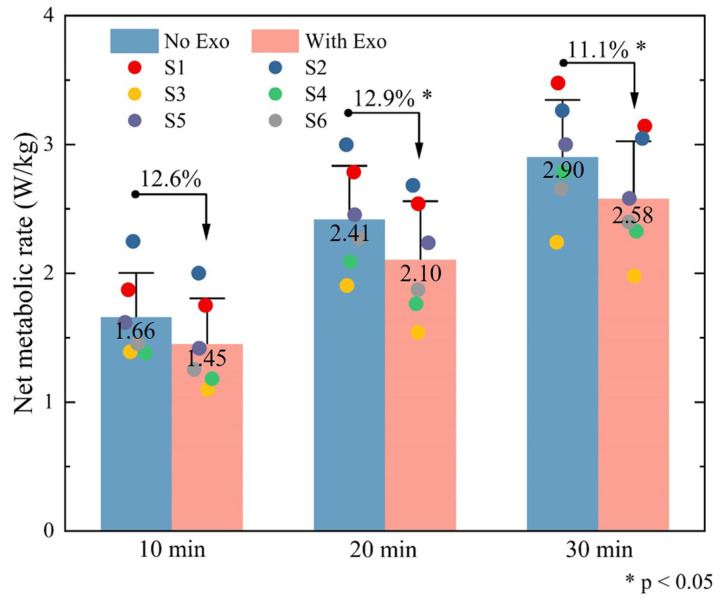
Metabolic performance in the first 10 min, first 20 min, and the entire duration, respectively. S1–S6 represent the data of 6 participants.

**Table 1 biomimetics-10-00349-t001:** Rotation angles of linkages and spinal forces in different spinal flexion.

α (°)	$\theta_{1}$ (rad)	$\theta_{2}$ (rad)	*T* (N)	$F_{N}$ (N)
30°	0.2239	0.4478	456.8564	531.2132
60°	0.6261	0.6261	665.7924	695.3729
90°	1.1762	0.5881	797.2055	809.2684

**Table 2 biomimetics-10-00349-t002:** Parameters of the five-bar linkage mechanism.

Linkage Number	Length (mm)	Original Angle ${(}^{\circ })$	Angle Range ${(}^{\circ })$
$L_{1}$	112	26	0
$L_{2}$	70	80	[−18.8, 51.4]
$L_{3}$	120	6	[−4.2, 25.8]
$L_{4}$	130	30	[−6.9, 29.1]
$L_{5}$	150	45	[−10, 20]

## Data Availability

The data used to support the findings of this study are available from the corresponding author upon reasonable request.

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
