# Peer review of "Development of a Passive Back-Support Exoskeleton Mimicking Human Spine Motion for Multi-Posture Assistance in Occupational Tasks"

_biomimetics, 2025, doi:10.3390/biomimetics10060349_

Round 1
Reviewer 1 Report
Comments and Suggestions for Authors
This paper presents a passive back-support exoskeleton assistance device, from concept proposal to design and experimental implementation. The work is promising in its potential to help reduce muscle fatigue during prolonged forward bending tasks. Overall, the quality of this paper is acceptable, with a well-organized structure and clearly presented results. However, there are some minor issues that should be addressed to further improve the quality of the article.
1. At the end of the Introduction section, please add a new paragraph—or revise the existing one—to provide a clearer overview of the content in the following sections.
2. In subsection 2.2.1, regarding the kinematic data collection, please clarify how many devices were used and their mounting positions with respect to the human body. Since data was collected from three trials, does the trajectory plotted in Figure 3 represent the average of all trials or a specific selection? Additionally, for the forces listed in Table 1, is there any method used to verify the accuracy of these values?
3. The statement, "To accommodate spinal motion patterns, the exoskeleton adopts a five-bar linkage mechanism to imitate the four natural curvatures of the human spine," found at the bottom of page 7, should be further elaborated. Please provide analytical reasoning for choosing a five-bar linkage over a four-bar or six-bar mechanism.
4. For the data presented in Table 2, please elaborate on how the key parameters for the exoskeleton linkage system were determined, and include relevant formulas if available. This information would be valuable for researchers and engineers seeking to replicate the system for users of varying heights and weights. Furthermore, to enhance the clarity of the design, please map and explain the corresponding links (L1–L5) and their lengths from Table 2 to Figure 7.
5. The trajectories in Figure 6 raise some concerns. The line plots appear very similar to those in Figure 3, which were collected using vision devices. If the intention is to demonstrate that the designed mechanism's end effector can replicate the observed motion, then the full trajectory of the five-bar linkage (including each joint) should be plotted and compared with the collected kinematic data.
6. The plot in Figure 9 needs clarification or revision. The light blue line starts at a value significantly lower than 30 degrees, based on the y-axis scale.
7. In the References section, there are some formatting issues and incomplete entries that need correction, including references [24], [26], [28], and [29].
Author Response
We sincerely appreciate your valuable feedback and insightful suggestions. Your thoughtful review has greatly contributed to improving the quality of our manuscript. We have made revisions based on your comments, with red strikethrough text indicating deleted content and blue text highlighting additions in the revised version. The responses to all of your comments are listed as follows:
Comment 1: At the end of the Introduction section, please add a new paragraph—or revise the existing one—to provide a clearer overview of the content in the following sections.
Response to Comment 1:
Thanks for your advice. We have added the paragraph in the end of the Introduction section and marked in blue.
Comment 2: In subsection 2.2.1, regarding the kinematic data collection, please clarify how many devices were used and their mounting positions with respect to the human body. Since data was collected from three trials, does the trajectory plotted in Figure 3 represent the average of all trials or a specific selection? Additionally, for the forces listed in Table 1, is there any method used to verify the accuracy of these values?
Response to Comment 2:
Thanks for your advice. We have revised subsection 2.2.1 accordingly.
- We have clarified that 24 Qualisys Arqus cameras were used for motion capture, along with the Qualisys Track Manager (QTM) software.
- 11 markers were placed on the spinous processes of 11 vertebrae: C7, T2,T4,T6,T7,T9,T12,L1,L3,L5, and S1.
- The trajectory presented in Figure 3 represents the average trajectory calculated across three trials.
The revised contents have been added in subsection 2.2.1 in Page 4 and highlighted in blue.
Although our modeling approach differs from those used in previous studies, the resulting trends in spinal compression and muscle force remain consistent with the literature. For example, both our model and prior studies (e.g., Schäfer, R., Trompeter, K., Fett, D. et al. The mechanical loading of the spine in physical activities. Eur Spine J 32, 2991–3001 (2023).) show that spinal loading and muscle forces increase progressively with greater trunk flexion angles. Similarly, when external loads are held by hand, spinal compression forces also increase accordingly.
While the absolute values may vary due to differences in model structure and assumptions, our primary objective was to capture these consistent biomechanical trends to demonstrate the necessity of designing a back-support exoskeleton. The observed increase in spinal and muscular loading under flexed postures and load handling tasks supports the rationale for mechanical assistance in such conditions.
Comment 3: The statement, "To accommodate spinal motion patterns, the exoskeleton adopts a five-bar linkage mechanism to imitate the four natural curvatures of the human spine," found at the bottom of page 7, should be further elaborated. Please provide analytical reasoning for choosing a five-bar linkage over a four-bar or six-bar mechanism.
Response to Comment 3:
Thanks for your comment. We have revised the manuscript to provide a more detailed explanation. The human spine features four distinct natural curvatures: cervical lordosis, thoracic kyphosis, lumbar lordosis, and sacral kyphosis. To closely fit the spine curvature, we choose a five-bar linkage mechanism. The five-bar linkage is constructed by selecting six key anatomical points: C7, T7, T12, L3, S1 and the pelvis (hip joint), connecting these points with rigid links. This arrangement directly maps each link to one of the spine’s four physiological curves:
C7-T7-T12 (Thoracic kyphosis):
These three vertebrae span the entire thoracic curvature. T7 serves as the apex of thoracic kyphosis.
T12-L3-S1 (Lumbar lordosis):
These three vertebrae span the entire lumbar curvature. L3 serves as the apex of lumbar lordosis, and S1 marking the transition to the sacrum.
S1-pelvis:
The pelvis is considered as a ‘pelvic vertebra’ or first vertebra of the spine. The pelvis functions as a fixed base analogous to the first sacral vertebra, setting the orientation of the entire lumbar links and aligning the spinal axis with the body’s center of gravity in the erect posture.
By tuning the five-bar linkage, the mechanism can replicate the shape of the spine during bending. A four-bar linkage mechanism only provides three independent joint and cannot mimic the spine movement. A six-bar linkage system could introduce an additional degree of freedom, it increases mass, mechanical complexity, and cable control burden, which could be influenced the wearable comfort and reliability. Thus, five-bar linkage offers an compromise: it captures the spinal curves with minimal added complexity, ensuring both biomechanical accuracy and mechanical simplicity.
We have added relevant explanations in the section 3.1 and highlighted in blue in Page 8.
Comment 4: For the data presented in Table 2, please elaborate on how the key parameters for the exoskeleton linkage system were determined, and include relevant formulas if available. This information would be valuable for researchers and engineers seeking to replicate the system for users of varying heights and weights. Furthermore, to enhance the clarity of the design, please map and explain the corresponding links (L1–L5) and their lengths from Table 2 to Figure 7.
Response to Comment 4:
Thanks for your advice. In the revised manuscript, we have added a detailed explanation of how link lengths and joint angle ranges in Table 2 were determined, as well as a mapping between back link (L1-L5) and its anatomical segment as shown in Figure 7.
The parameters were determined based on measurements of the human spine bending motion. The positions of the anatomical reference points including vertebrae C7, T7, T12, L3, S1 and the pelvis center were obtained using the motion capture system (Qualisys, Sweden). The distances between these points formed the basis for the five linkage segments, that is, L5: C7 to T7; L4: T7 to T12; L3: T12 to L3; L2: L3 to S1; L1: S1 to the pelvis center.
The initial joint angles were obtained when the subject is in the static standing posture. During flexion, changes in linkage angles were determined based on the dynamic variation in positions and intersegmental angles of these anatomical points, allowing the mechanism to mimic spinal motion throughout the bending process.
The relevant descriptions have been added in Section 3.1 in Page 9 and highlighted in blue.
Comment 5: The trajectories in Figure 6 raise some concerns. The line plots appear very similar to those in Figure 3, which were collected using vision devices. If the intention is to demonstrate that the designed mechanism's end effector can replicate the observed motion, then the full trajectory of the five-bar linkage (including each joint) should be plotted and compared with the collected kinematic data.
Response to Comment 5:
Thanks for your advice. We have revised Figure 6 to include the trajectories of all five joints in the five-bar linkage, and we overlaid them with the experimentally collected motion capture date of spine for direct comparison. This provides a clearer demonstration of the model’s ability to reproduce the observed human spinal curvature through the designed linkage.
Comment 6: The plot in Figure 9 needs clarification or revision. The light blue line starts at a value significantly lower than 30 degrees, based on the y-axis scale.
Response to Comment 6:
Thanks for your advice. We have redrawn Figure 9 and increase the y-axis resolution and annotate the initial value of each curve directly on the plot for precise interpretation.
Comment 7: In the References section, there are some formatting issues and incomplete entries that need correction, including references [24], [26], [28], and [29].
Response to Comment 7:
Thanks for your reminder. We have checked and updated the references.
Lastly, we hope the revisions meet your expectations and effectively address your concerns. Thank you again for your time and effort.

Reviewer 2 Report
Comments and Suggestions for Authors
I have reviewed the paper " A Passive Back-Support Exoskeleton Based on the Motion Mechanism of Human Spine for Multi-Posture Assistance" and found the paper can be accepted after major revision.
-More physical explanation of results is required.
-The Abstract should be improved.
-The quality of the figures should be improved.
-Based on the topic the title is so short and needs to be clarify the problem statement clearly.
-The Figures quality are too weak please improve the quality and put some arrays on the important part .
What materials are used in the construction of the exoskeleton, and how do they contribute to its overall weight, comfort, and effectiveness?
In what ways can the exoskeleton be adjusted or customized to accommodate different body types and activities for optimal support?
What are the preliminary findings regarding the exoskeleton's impact on reducing back pain or fatigue in users during prolonged activities?
-Finally, the language of the paper needs to be polished.
Author Response
We sincerely appreciate your valuable feedback and insightful suggestions. Your thoughtful review has greatly contributed to improving the quality of our manuscript. We have made revisions based on your comments, with red strikethrough text indicating deleted content and blue text highlighting additions in the revised version of our manuscript. The responses to all of your comments are listed as follows:
Comment 1: More physical explanation of results is required.
Response to Comment 1:
Thanks for your advice. We have revised the Discussions sections and the explanations have been added and highlighted in blue in Page 14.
Comment 2: The Abstract should be improved.
Response to Comment 2:
Thank you for the comment. We have revised the Abstract to clarify the methodology and compare the experimental results. The revised Abstract better reflects the structure of the study and highlighted in blue.
Comment 3: The quality of the figures should be improved.
Response to Comment 3:
Thanks for your advice. We have redrawn the figures and ensure the figures are in high resolution.
Comment 4: Based on the topic the title is so short and needs to be clarify the problem statement clearly.
Response to Comment 4:
Thanks for the insightful suggestion. We have revised the manuscript title to better reflect the problem addressed and the technical contribution of the study.
Comment 5: The Figures quality are too weak please improve the quality and put some arrays on the important part.
Response to Comment 5:
Thanks for your advice. We have redrawn the figures. Arrows and data have been added.
Comment 6:What materials are used in the construction of the exoskeleton, and how do they contribute to its overall weight, comfort, and effectiveness?
Response to Comment 6:
Thank you for the comment. We have added detailed descriptions of the materials used in the exoskeleton construction and discussed their impact on weight, comfort and performance in Section 4.1 in Page 12.
The exoskeleton frame was primarily constructed using aluminum alloy, selected for its high strength-to-weight ratio and good corrosion resistance, which ensures both structural integrity and reduced burden on the user. The back plate was made from 3D printing resin, with breathable mesh fabric attached, enhancing wearing comfort. The vertebral units were made with carbon fiber board and stainless steel to ensure mechanical durability under repetitive motion. The overall weight of the exoskeleton is approximately 3.2 kg.
Comment 7: In what ways can the exoskeleton be adjusted or customized to accommodate different body types and activities for optimal support?
Response to Comment 7:
Thank you for this important question. We have revised the manuscript to include a detailed explanation of the adjustability of the proposed exoskeleton, and provided in Section and highlighted in blue.
In the current version of the exoskeleton, basic size adjustments are achievable for users whose body dimensions fall within a typical range, that is, individuals who are neither significantly overweight nor underweight, and whose height variation is moderate. For such users, fine-tuning can be achieved through the telescopic adjustment of several key linkages.
Specifically, the lower limb support structure allows adjustment of linkage length between the joints to accommodate different thigh or calf lengths. As annotated in Figure 7, certain linkages are designed as telescopic rods, enabling incremental extension or retraction. The pulley steering structure at the pelvis includes the telescopic rods, enabling alignment with different pelvic widths. Similarly, for the back support structure, the linkages between vertebral units can be slightly adjusted in length to better conform to the natural curvature and posture of the individual user. Shoulder, thigh and calf straps are adjustable in both length and tension to ensure a secure yet comfortable fit.
However, for users with significantly different body types, such as those with notably lager or smaller body types, a standard adjustable version may not be suitable. In such cases, a customized exoskeleton is required. This can be achieved by following the design framework outlined in the paper, within includes redefining the key vertebral reference points and corresponding linkage dimensions.
We have added relevant descriptions in Discussion section and marked in blue in Page 15.
Comment 8: What are the preliminary findings regarding the exoskeleton's impact on reducing back pain or fatigue in users during prolonged activities?
Response to Comment 8:
Thank you for the comment.
In order to figure out the impact of exoskeleton on reducing back fatigue, experiments were conducted. Participants performed a 30-minute static bending assembly task under two conditions: with and without exoskeleton. To evaluate the erector spinae muscle activity, surface electromyography was recorded. Participants first performed five repeated bending motions before the 30-minute task, and EMG signals were recorded as the baseline. After completing the 30-minute assembly task, the same five bending motions were conducted again to record the erector spinae EMG signals. The EMG signals were all recorded without wearing the exoskeleton under two conditions. Metabolic cost using the measurement device (Cosmed, Italy) were used during the 30-minute task.
The EMG signals were post-processed using integrated EMG (IEMG) and Root mean square (RMS) analyses. The results revealed that, in the group wearing the exoskeleton during the task, IEMG values were reduced by 10.1%, and RMS values were reduced by 9.78% compared to the group without wearing the exoskeleton. The metabolic cost was reduced by 11.1% when using the exoskeleton for support. These results suggest that the exoskeleton helped reduce the burden on the erector spinae during prolonged trunk flexion. These reductions can be attributed to the ability of exoskeleton to share the lumbar extension torque, thereby lowering the need for sustained isometric contraction of the spinal extensors. Consequently, the exoskeleton can reduce the localized muscle fatigue, improves endurance and potentially mitigate the risk of developing chronic lower back discomfort.
Comment 9: Finally, the language of the paper needs to be polished.
Response to Comment 9:
Thank you for the comment. We have polished the language of the paper.
Lastly, we hope the revisions meet your expectations and effectively address your concerns. Thank you again for your time and effort.

Reviewer 3 Report
Comments and Suggestions for Authors
Dear authors,
I have the following concerns:
- abstract: you identify results, however you do not mention how have you compared the data.
- a brief description of the motion capture equipment should be provided.
- there is a huge difference regarding (for example) anthropommetrics and movement biomechanics between people. Do you think one person is enough to study the kinematics of the spine? Would it be similar for people with different age or with some spinal disorder? Or the exoskeleton is design for a specific small range of population?
- which is the main objective of studying the kinematic of spine bending? it should be clearly stated.
- which software was used to design the exoskeleton?
- For validation, you selected "six participants with similar body shapes". My questions is similar to the previous one: was the exoskeleton designed for a specific small range of population? Can you comment the benneficts/limitations of the approach? Would it be easily adapted to include other types of body shapes?
- Did you follow some approach such as user-centred design? Or do you consider to study the usability?
- "critical for optimizing performance and preventing fatigue." I suggest improve performance instead of optimizing.
Author Response
We sincerely appreciate your valuable feedback and insightful suggestions. Your thoughtful review has greatly contributed to improving the quality of our manuscript. We have made revisions based on your comments, with red strikethrough text indicating deleted content and blue text highlighting additions in the revised version. The responses to all of your comments are listed as follows:
Comment 1: abstract: you identify results, however you do not mention how have you compared the data.
Response to Comment 1:
Thanks for your comment. We have revised the Abstract and highlighted in blue. We added a brief explanation of the evaluation methods used to obtain and compare the results, including surface electromyography analysis and metabolic rate measurements. These comparisons were performed between the exoskeleton-assisted and non-assisted conditions during and after a 30-minute forward-bending assembly task.
Comment 2: a brief description of the motion capture equipment should be provided.
Response to Comment 2:
Thanks for your advice. We have added a brief description of the motion capture system used for kinematic data collection in Subsection 2.2.1. Specifically, we used a Qualisys (Sweden) motion capture system with 24 cameras to track the 3D trajectories of the reflective markers placed on key anatomical points of the participants’ spine and pelvis. The added description has been highlighted in blue.
Comment 3: there is a huge difference regarding (for example) anthropommetrics and movement biomechanics between people. Do you think one person is enough to study the kinematics of the spine? Would it be similar for people with different age or with some spinal disorder? Or the exoskeleton is design for a specific small range of population?
Response to Comment 3:
Thanks for your comment.
In our study, the proposed passive exoskeleton is intended for use in occupational environments such as assembly lines, construction sites, and operating rooms, where worker or medical professionals are required to maintain prolonged forward-bending postures. The target user group primarily consists of healthy adult workers, typically within the young to middle-aged adults, who are actively engaged in physically demanding tasks. The aim of the design is to provide back support and delay the onset of spinal fatigue or musculoskeletal disorders, rather than to serve as a rehabilitative device for individuals with existing spinal conditions.
Given this, individuals with severe spinal deformities or chronic back problems or elderly are generally not recommended to engage in such occupational setting, and thus fall outside the intended user population of our device. Therefore, the exoskeleton is not designed for these groups.
To study spinal kinematics for the design, we selected a representative healthy male subject with a height of 175 cm and standard body proportions. This choice reflects a standard anthropometric reference. The subject underwent repeated bending tests, from which we extracted stable motion trajectories of key vertebrae to inform the five-bar linkage structure. Additionally, we performed similar motion capture trials on other healthy individuals and observed consistent movement trends across participants, supporting the generalizability of the proposed mechanism within the targeted user range. Thus, we chose one result to display in the manuscript.
It is important to emphasize that the design methodology presented in this paper serves as a reference framework. Based on this approach, the exoskeleton can be further customized to accommodate individual body dimensions within the target population through adjustments to linkage lengths. And the present version of the exoskeleton has already applied the telescopic links for the users to adjust the length to fit the body shape.
We have added relevant descriptions in Discussion section and highlighted in blue.
Comment 4: which is the main objective of studying the kinematic of spine bending? it should be clearly stated.
Response to Comment 4:
Thanks for your comment. The primary objective of studying the kinematics of spine bending is to inform the mechanical design of a passive back-support exoskeleton that can effectively conform to human spinal motion and provide assistance across multiple postures.
Specifically, understanding the spatial and angular trajectories of key vertebrae during forward bending enables us to design a five-bar linkage mechanism that both accommodates natural spinal curvature changes and delivers targeted support at various flexion angles. This mechanism can ensure that the exoskeleton does not influence natural trunk movements while maintaining ergonomic alignment with the user’s back.
Moreover, the kinematic analysis allows for identifying the optimal placement of joints and links in the exoskeleton so that the structure can mimic the thoracolumbar curvature transition observed in human posture. Without such analysis, the risk of mechanical misalignment or user discomfort would significantly increase, undermining the efficacy of the exoskeleton in reducing muscle fatigue and supporting prolonged static tasks.
The relevant description has been added in Section 2.2.1.and marked in blue in Page 4.
Comment 5: which software was used to design the exoskeleton?
Response to Comment 5:
Thanks for your comment. The design and analysis of the exoskeleton system involved the integration of multiple software platforms. The 3D modeling and mechanical design of the exoskeleton were conducted using Solidworks 2020 (Dassault Systemes). To determine the key linkage parameters, including segment lengths and angular trajectories, we utilized MATLAB R2021a. For the acquisition of human spinal motion data, we used the Qualisys motion capture system equipped with 24 high-speed cameras, and processed the data using QTM (Qualisys Track Manager) software.
The relevant descriptions have been added in the Section 4.1 and highlighted in blue in Page 11.
Comment 6: For validation, you selected "six participants with similar body shapes". My questions is similar to the previous one: was the exoskeleton designed for a specific small range of population? Can you comment the benneficts/limitations of the approach? Would it be easily adapted to include other types of body shapes?
Response to Comment 6:
Thank you for the question. In this study, our primary design objective was to assist healthy workers, such as construction workers, doctors and field laborers, who are frequently required to maintain prolonged forward-bending postures in occupations such as manufacturing, surgery and agriculture. To ensure consistency and reduce variability in the experiments, we selected six participants (average height: 175cm, BMI within normal range, no known musculoskeletal disorders). The initial five-bar linkage parameters were derived from spinal kinematic data collected from individuals with these body characteristics. While this limits generalizability across all population types, it allows a focused assessment of the performance of the exoskeleton under specific conditions.
It is true that this version of the exoskeleton has limitations, particularly in adapting individuals with significantly different body shapes (e.g., very tall/ short, obese, or those with spinal disorders). However, the design method of the exoskeleton offers practical benefits in terms of scalability and customization. Specifically,
Adjustable link lengths: As shown in Figure 7, several connecting rods (both in the spine-mimicking structure and lower limb supports) are designed to be telescopic, allowing for minor body shape variations to be adapted through simple adjustments.
Custom fabrication potential: For users with significant anthropometric differences, the proposed design method can be reused to generate a customized exoskeleton configuration. The method involves capturing spinal motion trajectories (using motion capture system), and computing new linkage parameters using the same five-bar modeling logic.
In summary, the current design targets a standard user profile for occupational support applications. While initial testing was limited to a narrow population segment, the design methodology and structural adjustability provide a clear path for expanding applicability in future work.
We have added explanations and limitations in the Discussion section and marked in blue in Page 15.
Comment 7: Did you follow some approach such as user-centred design? Or do you consider to study the usability?
Response to Comment 7:
Thanks for your comment. Yes, in the design process of the back-support exoskeleton, we followed the principles of user-centered design to ensure the device meets the specific needs of target users such as construction workers or surgeons.
Specifically, we implemented the UCD approach as follows:
User Needs Assessment:
We first discussed with relevant workers to understand their typical tasks, postural challenges and pain positions related to prolonged standing or bending. This help us identify key design requirements such as posture stabilization, lumbar support.
Design and Ergonomic Optimization:
Based on the needs of the user, the mechanical structure was designed to mimic human spine curvature and flexibility while avoiding interference with normal range of motion. Five-bar linkage mechanism were key design constraints. Meanwhile. according to the needs of users, the lightweight materials and passive support mechanisms were chosen.
Prototype evaluation:
We invited potential users to wear the prototype and perform assembly tasks. We collected quantitative data (e.g., EMG measurements, metabolic cost). This evaluation guided refinements to the structure of exoskeleton. Overall, participants reported reduced fatigue and acceptable wearing comfort during the 30-minute task. These results are valuable for guiding future structure iterations.
We fully recognize the importance of systematic usability evaluation in exoskeleton development. In future work, we will continuously conduct usability studies, including both quantitative scales (e.g., comfort rating) and task performance, to comprehensively evaluate long-term wearability and user satisfaction in real industrial scenarios.
Comment 8: “critical for optimizing performance and preventing fatigue." I suggest improve performance instead of optimizing.
Response to Comment 8:
Thanks for your advice. We have revised the descriptions and highlighted in blue in Page 14.
Lastly, we hope the revisions meet your expectations and effectively address your concerns. Thank you again for your time and effort.

Round 2
Reviewer 2 Report
Comments and Suggestions for Authors
It can be accepted int his form
Reviewer 3 Report
Comments and Suggestions for Authors
Dear authors,
I appreciate you have addressed m concerns.